# A Nectin1 Mutant Mouse Model Is Resistant to Pseudorabies Virus Infection

**DOI:** 10.3390/v14050874

**Published:** 2022-04-22

**Authors:** Xiaohui Yang, Chuanzhao Yu, Qiuyan Zhang, Linjun Hong, Ting Gu, Enqin Zheng, Zheng Xu, Zicong Li, Changxu Song, Gengyuan Cai, Zhenfang Wu, Huaqiang Yang

**Affiliations:** 1National Engineering Research Center for Breeding Swine Industry, College of Animal Science, South China Agricultural University, Guangzhou 510642, China; yangxiaohui@stu.scau.edu.cn (X.Y.); yuchuanzhao@stu.scau.edu.cn (C.Y.); qiuyan@stu.scau.edu.cn (Q.Z.); linjun.hong@scau.edu.cn (L.H.); tinggu@scau.edu.cn (T.G.); eqzheng@scau.edu.cn (E.Z.); stonezen@scau.edu.cn (Z.X.); lizicong@scau.edu.cn (Z.L.); cxsong@scau.edu.cn (C.S.); cgy0415@scau.edu.cn (G.C.); 2Guangdong Provincial Key Laboratory of Agro-Animal Genomics and Molecular Breeding, South China Agricultural University, Guangzhou 510642, China

**Keywords:** antiviral breeding, genetic modification, nectin1, pig, PRV, disease resistance

## Abstract

The present study generated nectin1-mutant mice with single amino acid substitution and tested the anti-pseudorabies virus (PRV) ability of the mutant mice, with the aim to establish a model for PRV-resistant livestock. A phenylalanine to alanine transition at position 129 (F129A) of nectin1 was introduced into the mouse genome to generate nectin1 (F129A) mutant mice. The mutant mice were infected with a field-isolated highly virulent PRV strain by subcutaneous injection of virus. We found that the homozygous mutant mice had significantly alleviated disease manifestations and decreased death rate and viral loading in serum and tissue compared with heterozygous mutant and wild-type mice. In addition to disease resistance, the homozygous mutant mice showed a defect in eye development, indicating the side effect on animals by only one amino acid substitution in nectin1. Results demonstrate that gene modification in nectin1 is an effective approach to confer PRV resistance on animals, but the mutagenesis pattern requires further investigation to increase viral resistance without negative effect on animal development.

## 1. Introduction

Pseudorabies virus (PRV) is an economically important pathogen causing severe losses to pig production. PRV causes reproductive and respiratory problems in breeding and finishing pigs and central nervous system signs and high mortality in piglets [1,2,3,4]. Although the natural reservoir is pig, PRV has a broad host range and can lethally infect a wide variety of animal species [5,6,7,8,9]. PRV infection in pigs is controlled using inactivated and attenuated live vaccines [10]. However, PRV outbreak in vaccinated pig populations still occur widely in many countries mainly due to variations of virus, usually compromising vaccine effectiveness and increasing virulence to pigs [1,11,12].

PRV belongs to *herpesvirinae* family and *alphaherpesvirinae* subfamily [13,14]. The representative alphaherpesviruses, such as PRV and herpes simplex virus (HSV), share some common molecular machinery and mechanism to infect host cells [15,16]. The host membrane-bound protein, nectin1, plays a pivotal role in alphaherpesvirus infection. Global nectin1 gene knockout in mice can confer resistance to HSV1 and HSV2 infection, resulting in a milder disease after HSV1 and HSV2 infection in mice [17,18,19,20,21]. Our previous work found that nectin1 knockout in pig cells reduce PRV growth by impairing cell-to-cell spread step of virus [22]. In this respect, nectin1 gene modification in animals provides a practical route for PRV antiviral breeding given that genetic engineering approach can be used for pig breeding.

We established a mouse model carrying specific mutation in nectin1 to investigate the anti-PRV effect of nectin1 gene modification in animals. Previous work reported that the extracellular N-terminal variable region-like (V) domain of nectin1, which interacts with viral glycoprotein D (gD), is important for alphaherpesvirus infection. The critical residues in V domain for PRV-gD/nectin1 engagement include N77, I80, M85, R110, and F129 [23,24]. Amino acid substitution in some of the residues can severely reduce PRV-gD/nectin1 binding and PRV entry activity. The in vitro cell-based results showed that mutation in F129 has fewer side effects than other key residues on homotypic and heterotypic nectin–nectin interactions, which are important for cell adhesion [23]. Considering these factors, we introduced an F129A single amino acid substitution in mouse nectin1 gene. The homozygous F129A mutant mice were studied with respect to their infection levels in PRV challenge experiments. We also investigated the impact of mutation on the physiologic function of nectin1 in vivo.

## 2. Materials and Methods

### 2.1. Animals

The animals and procedures in this study were in accordance with the guidelines and approval of the Institutional Animal Care and Use Committees at South China Agricultural University (approval ID: 2020B032). We used CRISPR-mediated homology-directed repair method for amino acid substitution in mouse genome. The guide RNA (gRNA) sequence targeting mouse nectin1 gene was ACGGTTGCCCGTAGGGAAGG, and a donor oligo containing F129A (TTC to GCT) mutation was designed as follows: ATCCGCCTCTCCGGTCTGGAGCTGGAGGACGAGGGCATGTACATCTGTGAATTTGCCACCGCTCCTACGGGCAACCGTGAAAGCCAGCTCAATCTCACTGTGATGGGTAAGCTGCCCTGGGCC; the mutation site is underlined. The gRNA, donor oligo, and Cas9 mRNA were co-injected into fertilized eggs of C57BL/6 mice to generate targeted knock-in offspring. F1 founder animals were identified by PCR followed by sequence analysis, and then bred to test germline transmission and generate F2 mutant animals.

### 2.2. Analysis of Nectin1 Expression

Mouse brains were minced and lysed in Pierce RIPA buffer (Thermo Fisher Scientific, Rockford, IL, USA) supplemented with protease inhibitor cocktail (Thermo Fisher Scientific, Rockford, IL, USA). Tissue lysate was collected by centrifugation and quantified using Pierce BCA protein assay kit (Thermo Fisher Scientific, Rockford, IL, USA). An equal amount of lysate was boiled in loading buffer and subjected to SDS-PAGE. Subsequently, the separated protein in gel was transferred onto the PVDF membrane, which was then blocked in 5% milk and incubated with anti-nectin1 mouse monoclonal antibody (sc-21722, Santa Cruz Biotechnology, Dallas, TX, USA) at 4 °C overnight. The membrane was washed and further incubated with HRP-conjugated goat-mouse IgG secondary antibody. After thorough washing, the target protein was imaged using a SuperSignal West Pico enhanced chemiluminescence kit (Thermo Fisher Scientific, Carlsbad, CA, USA). The same blots were probed with a β-actin rabbit monoclonal antibody (13E5, Cell Signaling Technology, Danvers, MA, USA) as a loading control.

### 2.3. Viral Culture and Titration

A field-isolated PRV strain was propagated and titered in PK15 cells [22]. Virus diluted in DMEM was inoculated in PK15 cell monolayer and cultured to reach full cytopathic effect (CPE) in 5% CO_2_ incubator at 37 °C. The culture was collected and aliquoted as viral stock. To examine the viral titers, TCID50 assay was performed on PK15 cells grown in 96-well plates. Virus was serially diluted from 10^−^^1^ to 10^−^^10^ in DMEM, and 0.1 mL of viral dilution was added per well; eight wells were infected per dilution. Virus was allowed to adsorb to cells for 2 h, followed by replacing the viral dilution with 0.1 mL of fresh DMEM for each well. The plates were cultured in 5% CO_2_ incubator at 37 °C for monitoring CPE for one week. The titer was calculated using the method of Muench and Reed [25].

### 2.4. Viral Infection in Mice

Homozygous mutant mice and age-matched controls (wild-type (WT) and heterozygous) were injected subcutaneously in the neck at a single dose of 100 μL virus prediluted to the indicated titers (10^5.43^ and 10^7.42^ TCID50 for two viral challenge experiments). Signs of disease and survival rate of infected mice were recorded for 96 h. The symptoms of PRV infection were scored using a 3-point system: 0 = normal posture; 1 = attempt to scratch or slight scratching; 2 = frequent scratching and abnormal posture such as hunchback; and 3 = scratching with biting and bleeding of the wound. All mice were sacrificed at 96 h to collect serum and various tissues for viral titer quantification. Viral DNA in serum and brain tissues in the same amount was extracted with RaPure Viral RNA/DNA Kit (Magen, Guangzhou, China) and subjected to quantitative PCR (qPCR) by using Premix ExTaq (Probe qPCR) (Takara, Dalian, China) and the following primers and probe, PRV-gE-Forward, CCCACCGCCACAAAGAACACG, PRV-gE-Reverse, GATGGGCATCGGCGACTACCTG, and PRV-gE-Probe, FAM-CAGCGCGAGCCGCCCATCGTCAC-BHQ1 in a QuantStudio 7 Flex Real-Time PCR System (Thermo Fisher Scientific, Foster City, CA, USA). A PRV gE gene plasmid was serially diluted as templates to generate a reference curve between Ct value and DNA copies in the same qPCR reaction of viral DNA samples. Viral DNA was quantified using the standard curve and expressed as genome equivalents (GE) in extracted DNA solution.

### 2.5. Histological Analysis

The eyes of mice were collected and soaked in FAS eye fixative (Servicebio, Wuhan, China) for 24 h at room temperature. The fixed tissues were dehydrated with gradient concentrations of ethanol, embedded in paraffin wax, and sectioned to tissue slice of 5 μm thickness. For Hematoxylin and Eosin (H&E) staining, the tissue sections were deparaffinized in the xylene and rehydrated by passing through decreasing concentrations of ethanol baths and water. The rehydrated tissue sections were stained in hematoxylin for 5 min at room temperature. The sections were rinsed in tap water and differentiated in 1% HCl in 70% alcohol for 5 min. After rinsing in tap water, the sections were treated with ammonia water to convert the hematoxylin to a dark blue color. The sections were then rinsed and stained in 1% Eosin Y for 10 min at room temperature. After staining, the sections were washed in tap water for 5 min, dehydrated in increasing concentrations of ethanol, cleared in xylene, and mounted in mounting media for microscopy assay.

### 2.6. Statistical Analysis

All statistical analyses were performed using Prism (GraphPad, v8, LaJolla, CA, USA). Mean ± standard deviation was calculated for replicate data. Means comparisons in body weight and viral copies were conducted using unpaired *t* tests or ANOVA, and survival rates were compared with Gehan–Breslow–Wilcoxon test, as indicated in the figure legends.

## 3. Results

### 3.1. Generation of Nectin1 (F129A) Mutant Mice

Nectin1 F129 residue is conserved across different mammalian species (Appendix A). Its functional significance for engagement of gD of multiple alphaherpesviruses has been found in nectin1 of mouse, pig, and cattle [23,24,26]. Nectin1 (F129A) mutant mice were produced by CRISPR-mediated knock-in using zygote injection of CRISPR system cleaving nectin1 and a DNA template donor carrying F129A mutation (TTC→GCT in DNA sequence) (Figure 1A). The founder mice were genotyped by PCR amplification of nectin1 target region and Sanger sequencing identifying the presence of mutation. The positive mice were bred to generate homozygous and heterozygous mutant offspring (Figure 1B). To detect if the nectin1 protein expression was affected by amino acid substitution, the brain tissues from WT, heterozygous, and homozygous mice were lysed to extract total protein. Western blot assay showed that all three genotypes of mice had nectin1 expression in the similar levels, indicating no impact of F129A mutation on protein expression in brains (Figure 1C). The homozygous mutant mice generally had similar growth rate to WT and heterozygous mutants within one month after birth. The increase in body weight of homozygous mutants was slightly slower than WT and heterozygous mutants after one month, and the difference was more significant in female mice (Figure 1D,E).

### 3.2. Anti-PRV Ability of Nectin1 (F129A) Mutant Mice

We conducted two viral challenge experiments to test the PRV infection status in the mutant mice. Prior to experiments, we evaluated the susceptibility of the mouse strain we used (C57BL/6) to PRV. Our pre-experiments showed that the attenuated PRV vaccine strain Bartha-K61 was mildly pathogenic to C57BL/6 mice, but a field-isolated highly virulent strain of PRV [22] can cause acute symptoms with a high mortality in 3–10 days after viral inoculation in different doses. Infected mice showed viremia and tissue lesion, and high viral load in infected tissues can be detected by qPCR (data not shown). The susceptibility of C57BL/6 mice to PRV has also been comprehensively investigated in previous reports [27,28,29]. Upon confirming the PRV susceptibility of mice, we performed experiment 1, in which seven WT and seven homozygous at six weeks were given subcutaneous injection with 1 × 10^5.43^ TCID50 PRV field strain (Figure 2A). After PRV inoculation, three mice of each genotype were sacrificed to collected tissues and serum to detect tissue morphology and viral loads at 36 h. The four other mice in each group were maintained to observe disease development for 96 h. Among them, two WT mice had symptoms of scratching and body incoordination starting at 48 h after inoculation and died at 72 h and 96 h, respectively (Figure 2B,C and Appendix A). All WT mice displayed acute itch symptom during the challenge period, but only two homozygous mutant mice had slight symptom starting at 84 h (Appendix A). qPCR results showed that viral copies in brain and serum of homozygous mice were lower than WT (Figure 2D).

We further repeated viral challenge experiment (experiment 2) by using a high dose of virus and including heterozygous mice as subjects. A total of nine WT, nine heterozygous, and seven homozygous at the age of eight weeks were subcutaneously injected with 1 × 10^7.42^ TCID50 PRV field strain (Figure 3A). Follow-up observations showed that six WT and seven heterozygous mice displayed neurological symptoms, such as scratching and biting, starting from 36 h, and finally died within 96 h. In the homozygous group, one showed neurological symptoms at 72 h and died at 96 h; three had slight scratching behavior at 96 h (Figure 3B and Appendix A). All infected mice were sacrificed at 96 h for serum and tissue sampling. Viral DNA extracted from half brain and serum in the same amount showed a significantly reduced viral copies in homozygous mice compared with WT and heterozygous. The viral copies in serum and brain between WT and heterozygous were not different significantly (Figure 3C).

### 3.3. Defect in Eye Development in Nectin1 (F129A) Mutant Mice

We noted an abnormal eye development in nectin1 (F129A) mutant mice. Homozygous mutant mice showed microphthalmia (Figure 4A). Histological analysis of the eyes showed severe deformation of eye structure in homozygous mice. The vitreous body totally disappeared, and the lenses adhered to the ciliary epithelia and retinal layers (Figure 4B). Furthermore, we observed a deformed ciliary body, which loses ciliary processes. Ciliary processes include the double epithelial layer consisting of pigment and non-pigment epithelia. As shown in Figure 4C, WT mice had the ciliary processes with the contacted pigment and non-pigment epithelia. However, such double-layer structure was not observed in ciliary body of homozygous mutant mice, which thus cannot make the ciliary processes (Figure 4C). A similar defect in eye development was also found in global nectin1 knockout mice or transgenic mice expressing the first V domain of nectin1, in which the main feature is microphthalmia with disappeared vitreous body and ciliary processes in eyes [30,31]. Western blot assay showed that nectin1 protein level in eyes of homozygous mutant mice was greatly less than that in WT littermates, however, nectin1 protein level did not differ greatly in brains between them (Figure 4D). Severe eye abnormality may be partly explained by specifically reduced nectin1 expression in eyes in homozygous mutant mice.

## 4. Discussion

### 4.1. Nectin1 Modification Resists PRV Infection in Hosts

As a representative member of alphaherpesvirus, PRV prevails in pig population and remains a serious threat to the current pig industry in many countries. The emerging animal genetic engineering technology provides an alternative strategy to modify host gene instead of treating the virus itself for viral prevention and control in pigs. The host gene modification antiviral strategy has been proven effective to control viral infection in animals, offering a simple pathway for antiviral breeding to benefit livestock production [32,33]. Nectin1 or nectin2 have long been recognized as the key host factor mediating infection of alphaherpesviruses [15,16]. Although a similar mode by interaction of viral gD and host nectin1 is proposed for entry and spread of alphaherpesviruses, different amino acids in gD/nectin1 binding interface are utilized for nectin1 engagement for different alphaherpesviruses. Structural data of pig nectin1-bound PRV gD revealed N77, I80, M85, and F129 of pig nectin1 as the key residues for their contacting. A mutagenesis study of the key interface residues in nectin1 further confirmed the functional necessity of F129 in PRV-gD engagement [23,24]. Our in vivo viral challenge results were in agreement with the in vitro structural and functional data, that is, homozygous F129A mutant mice harbored enhanced resistance to PRV infection, manifested by reduced viral load in serum and brain, alleviated tissue lesion, and enhanced survival rate compared with WT and heterozygous mutant mice. The mice model-based results lay the groundwork for anti-PRV study in pigs by genetic engineering approaches.

### 4.2. Correlation of PRV Infection in Mice and Pigs

PRV can infect a wide range of host animals. Similar host factors and machinery are employed by the virus for cell entry and replication in hosts. As a common coreceptor for multiple human and animal alphaherpesviruses, nectin1 may serve a similar role for alphaherpesvirus infection in both natural and non-natural hosts [16]. Therefore, the mouse model of PRV infection could be a valuable reference to study virus–host interaction in pigs. Some previous publications have also reported using transgenic mice as a model for pseudorabies-resistant livestock [34,35]. However, mice and pigs display different clinical manifestations after PRV infection [27,28,29]. PRV always causes a severe neuropathic itch followed by acute death. The virus replicates in the skin and peripheral nervous system (PNS) neurons, but few infections exist in the brain. In adult pigs, PRV infects mucosal epithelium and spreads to PNS neurons, where a quiescent, latent infection is established. Viral replication can be re-activated to spread back to mucosal surfaces, mainly causing respiratory disease. Acute itch and death rarely occur in adult pigs [27,28,29]. Investigations indicate that different immune responses, such as type I IFN or specific inflammation, mainly contribute to the difference in pathogenesis and clinical outcomes of PRV infection between mice and pigs [28,29]. The immunity-controlled differential disease manifestations could not affect the disease-resistance phenotype conferred by nectin1 modification, which is a host factor controlling PRV entry or spread in natural and non-natural hosts. Nectin1-mutation could be a universal anti-PRV even anti-alphaherpesvirus strategy in susceptible hosts.

### 4.3. Nectin1 (F129A) Mutation Impairs Nectin1 Physiologic Function

A previous work reported presence of deleterious effects of global nectin1 knockout on eye development in mice. Thus, we used single amino acid substitution other than global knockout for anti-PRV mouse preparation. However, only single amino acid mutation seemed to severely impair the normal physiological role of nectin1 in mice. A similar impairment to eye development to that in global nectin1 knockout mice was observed in F129A mutant mice [30]. Our result provides in vivo evidence that F129 is the residue critical for nectin–nectin homophilic or heterophilic trans-interactions; it is essential to form a functional ciliary body with double cell layer structure. This result also implies that PRV infection and nectin1-mediated cell adhesion utilize the same molecular area or machinery to exert their functions. Further work needs to precisely identify the different regions/residues of nectin1 separately hijacked by alphaherpesviruses for cell entry and working in homophilic or heterophilic trans-interactions to maintain normal cellular functions.

### 4.4. Exploration of Ideal Anti-PRV Gene Targets with Breeding Values in Pigs

The impaired eye function and other undefined abnormality caused by nectin1 mutation could affect the application of the same gene modification for pig antiviral breeding. The genetic engineering technology in agricultural animals should be safely utilized to confer desired phenotypes without penalties to economic traits or animal welfare. The ideal anti-PRV gene targets can be exploited by identifying the safe nectin mutant genotypes or other host factors essential for PRV infection. First, more cell-based works should be performed to define the specific regions in nectin1 or 2 that are critical for viral infection, but minimally interfering with normal cellular functions. Second, other reported gene targets, such as PVR, which may play a key role in PRV cell entry, can be investigated in animals with respect their antiviral ability in vivo [16,36]. Third, the recently developed CRISPR library can effectively screen host genes essential for viral infection [37,38]. A genome-wide screen has been performed for some alphaherpesviruses including PRV and Bovine Herpes Virus Type 1 (BHV-1) and revealed more potential gene targets, which are promising resources facilitating antiviral study [39,40].

## Figures and Tables

**Figure 1 viruses-14-00874-f001:**
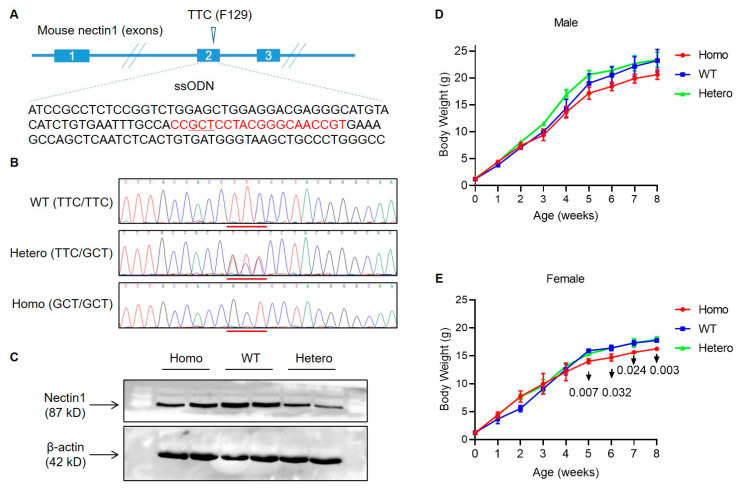
Characterization of nectin1 (F129A) mutant mice. (**A**) Gene targeting strategy to introduce F129A mutation in mouse nectin1. Sequences in red letters are gRNA-recognized target harboring F129A mutation (underlined sequences). (**B**) Sequences of nectin1 mutation site in homozygous, heterozygous, and WT mice. The underlined sequences are F129 mutation sites. (**C**) Nectin1 expression status in brain tissues of homozygous, heterozygous, and WT mice. (**D**,**E**) Growth curves of male (**D**) and female (**E**) F129A mutant mice. Female homozygous mutant mice had significantly reduced body weight compared with WT littermates at five, six, seven, and eight weeks. Data are presented in mean ± standard deviation. *p*-values represent homozygous versus WT and were analyzed with one-way ANOVA followed by Dunnett’s multiple comparison test. ssODN, single-stranded oligodeoxynucleotides; Homo, homozygous nectin1 (F129A) mutant mice; Hetero, heterozygous nectin1 (F129A) mutant mice; WT, wild-type mice.

**Figure 2 viruses-14-00874-f002:**
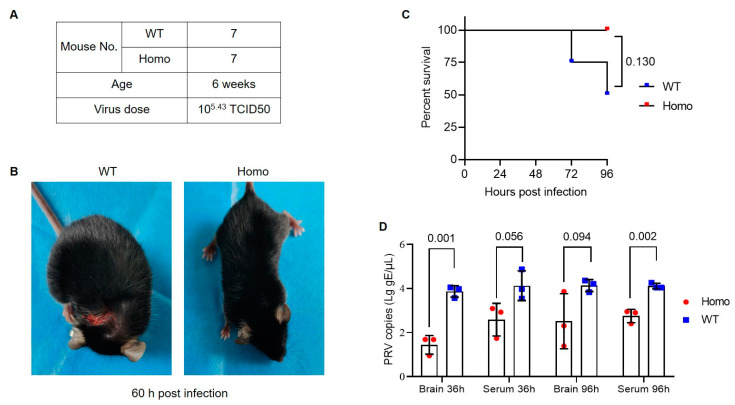
Viral challenge experiment 1 in nectin1 (F129A) mutant mice. (**A**) Experimental groups and virus dose for PRV challenge in mice. (**B**) Typical symptoms present in PRV-infected mice. A WT mouse showed body incoordination and severe wound in the neck because of scratching of the viral injection site at 60 h after challenge (left), whereas no symptoms were observed in nectin1 (F129A) homozygous mutant mouse (right). (**C**) Survival curve for nectin1 (F129A) homozygous mutant and WT mice after PRV challenge within 96 h. Four mice in each group were used for analysis of survival curve. The other three mice in each group were sacrificed to analyze viral infection level in various tissues at 36 h. (**D**) PRV viral loads at 36 h and 96 h in brain and serum by qPCR quantification of PRV gE gene. All data are presented in mean ± standard deviation. Statistically significant differences between mutant and WT groups are indicated by *p*-values, analyzed with Gehan–Breslow–Wilcoxon test and unpaired *t*-test for survival curve and viral copies, respectively. Homo, homozygous nectin1 (F129A) mutant mice; WT, wild-type mice.

**Figure 3 viruses-14-00874-f003:**
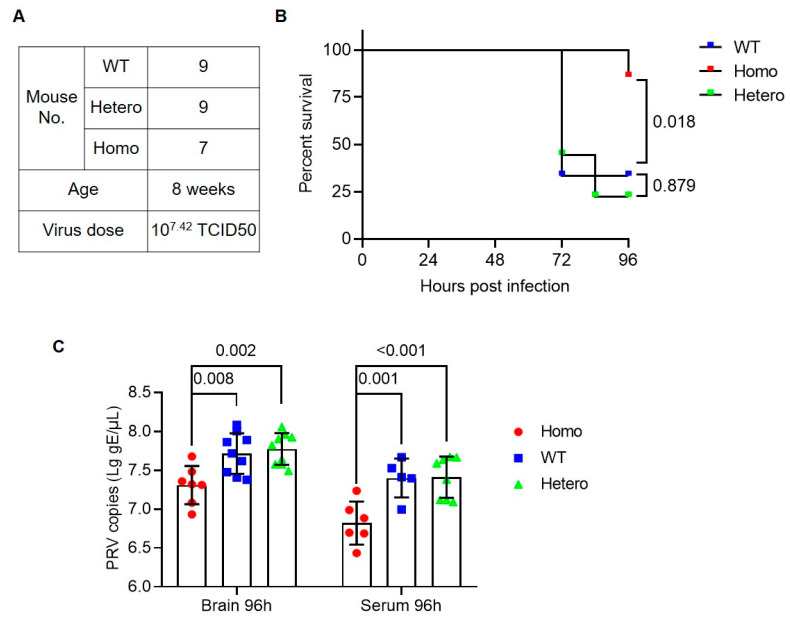
Viral challenge experiment 2 in nectin1 (F129A) mutant mice. (**A**) Experimental groups and virus dose for PRV challenge. (**B**) Survival curve for nectin1 (F129A) homozygous mutant, heterozygous mutant, and WT mice after PRV challenge within 96 h. (**C**) PRV viral loads in brain and serum by qPCR quantification of PRV gE gene. All surviving mice were sacrificed to analyze viral load in brains and serum at 96 h after viral challenge. Data are presented in mean ± standard deviation. Statistically significant differences are indicated by *p*-values, analyzed with Gehan–Breslow–Wilcoxon test and two-way ANOVA for survival curve and viral copies, respectively. Homo, homozygous nectin1 (F129A) mutant mice; Hetero, heterozygous nectin1 (F129A) mutant mice; WT, wild-type mice.

**Figure 4 viruses-14-00874-f004:**
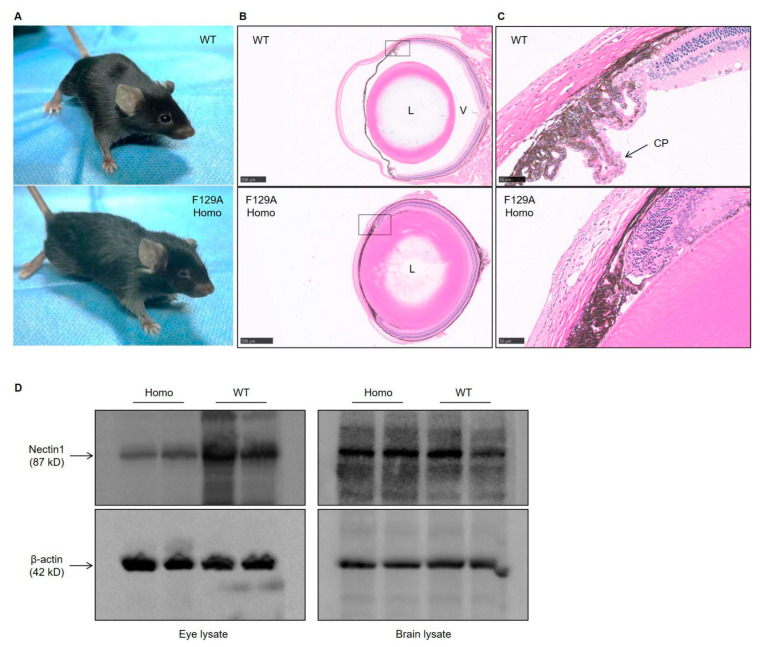
Defect in eye development in nectin1 (F129A) mutant mice. (**A**) Nectin1 (F129A) homozygous mutant mice show bilateral microphthalmia. Representative images of the eyes of nectin1 (F129A) homozygous mutant and age-matched WT mice are shown. (**B**) Histological analysis of the eyes of nectin1 (F129A) homozygous mutant and age-matched WT mice. The absent vitreous body and abnormal lenses can be found in homozygous mutant mice. (**C**) Magnified views of the ciliary body in the boxed areas in the panel (**B**). The ciliary body of WT mice displays the double cell layer structure of the ciliary epithelia composed of pigment (black-colored cells) and non-pigment cells (stained only by eosin in cytoplasm), whereas the homozygous mutant mice have deformed ciliary body, in which the ciliary processes with double cell layer structure of the ciliary epithelia are absent. (**D**) Nectin1 expression levels in eyes and brains of homozygous mutant and WT mice. Tissues of two mice from the same litter for each group were used for western blot assay. Homo, homozygous nectin1 (F129A) mutant mice; WT, wild-type mice; L, lens; V, vitreous body; CP, ciliary processes. Scale bars: 500 μm in (**B**), 50 μm in (**C**).

## Data Availability

All original data are available upon request.

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
