# Peer review of "A Nectin1 Mutant Mouse Model Is Resistant to Pseudorabies Virus Infection"

_viruses, 2022, doi:10.3390/v14050874_

Round 1
Reviewer 1 Report
In this manuscript, Yang, et al. established a nectin1-mutant mouse model with single amino acid substitution and tested anti-pseudorabies virus (PRV) ability of the mutant mice. Although the homozygous mutant mice showed a defect in eye development, the mice model-based results lay the groundwork for anti-PRV study in pigs by genetic engineering approaches. In addition, the manuscript is well written, and logically presented. However, I would like to suggest that the authors consider the following comments.
- Did the authors do HE and IHC to compare histopathological changes and viral load in nectin1 (F129A)-mutant mice with WT mice? I would like to suggest the authors to provide the results if HE and IHC tests were performed, as I note that the histopathological changes of the eyes were provided in results 3.3.
- Please proofread the manuscript carefully. For example: in line 95, “PRV strainwas…” should be “PRV strain was…”.
Author Response
- Did the authors do HE and IHC to compare histopathological changes and viral load in nectin1 (F129A)-mutant mice with WT mice? I would like to suggest the authors to provide the results if HE and IHC tests were performed, as I note that the histopathological changes of the eyes were provided in results 3.3.
Thank you for your valuable comments. Actually, we have conducted IHC of PRV antigen in brain cortex of infected mice. However, very few positive signals can be detected in the cortex of infected mice, thus the difference in PRV antigen is not obvious in brain slices between homozygous mutant and WT mice. Through checking literatures, we can find that mice and pigs display different clinical manifestations after PRV infection. PRV always causes a severe neuropathic itch followed by acute death. The virus replicates in the skin and peripheral nervous system (PNS) neurons, but few infections exist in the brain. In adult pigs, PRV infects mucosal epithelium and spreads to PNS neurons, where a quiescent, latent infection is established. Viral replication can be re-activated to spread back to mucosal surfaces, mainly causing the respiratory disease. Acute itch and death rarely occur in adult pigs. These differences are mainly caused by different immune response following PRV infection in native and nonnative hosts. We have add a paragraph in section 4.2 to discuss this point.
- Please proofread the manuscript carefully. For example: in line 95, “PRV strainwas…” should be “PRV strain was…”.
Thank you for pointing this mistake. We have revised it in resubmitted manuscript. We also carefully check language throughout the manuscript to minimize the typos and grammatical mistakes.
Reviewer 2 Report
This manuscript describes on Nectin-1 mutant mouse resistance against PRV infection.
Here are my comments:
- To prove this conclusion, authors should provide whether backbone mice are susceptible to PRV or not. Which backbone mice authors used and prove they are susceptible. Is it C57BL/6?
- How we either conclude or correlate these results with PRV infection in pigs?
- How much similarity of nectin 1 protein between mice and pigs?
- In Fig 1, authors shown very big difference in body weight. One amino acid difference between homo and hetero induced very big difference in body weight, what could be the reason?
- In Fig 2, Homo mice were more resistant than WT mice which is different results from results on Fig 1. How you can explain?
- In Fig 3, viral loads on homo and hetero were more or less same but very big different in survival rate. What could be the reason?
Author Response
- To prove this conclusion, authors should provide whether backbone mice are susceptible to PRV or not. Which backbone mice authors used and prove they are susceptible. Is it C57BL/6?
Yes. The backbone of tested mice are C57BL/6. C57BL/6 is lethally susceptible to PRV infection. Some previous reports have clearly studied this. We have added the background information of mice and some references in Sections 2.1 and 3.2 in the revised manuscript.
- How we either conclude or correlate these results with PRV infection in pigs?
Mouse models provide useful references for pigs infection with PRV. PRV can lethally infect many animals with a broad host range, implying that PRV can engage the host factors including nectin1 from different species for effective viral infection. To address the reviewers’ concern, (1) we added a sequence alignment of nectin1 of mice and pigs to show their conservation (Figure S1); (2) we added some references reporting using mice to establish PRV resistant animal models (references 34 and 35); (3) we added a paragraph in Discussion section to discuss the correlation of mice and pigs in PRV infection (Section 4.2).
- How much similarity of nectin 1 protein between mice and pigs?
They have a high similarity with 93.8% homology in protein sequence. We added the nectin1 sequence alignment results among multiple species in Figure S1.
- In Fig 1, authors shown very big difference in body weight. One amino acid difference between homo and hetero induced very big difference in body weight, what could be the reason?
Yes. One amino acid substitution in nectin1 can result in significant abnormality in animals. The major abnormality is eye defect in homozygous mutant mice. In addition, such mutation may impair other tissues as nectin1 works to adhere cell to cell. We have studied the eye histology of mutant mice in section 3.3. The functional impairment of nectin1 by F129A mutation thus results in the slightly reduced body weight gain of homozygous mutant mice.
- In Fig 2, Homo mice were more resistant than WT mice which is different results from results on Fig 1. How you can explain?
Our major conclusion is that homo mice is resistant to PRV. This result is consistent throughout the manuscript. Figure 1 shows the growth status of mice, not the disease infection result. The homo mice grow up slightly slowly, but they resist PRV infection.
- In Fig 3, viral loads on homo and hetero were more or less same but very big different in survival rate. What could be the reason?
Thank you for your comments. Probably the figure is not marked clearly enough. The WT and hetero are similar in viral load and survival rate. Homo mice has low viral load and high survival rate, compared to WT and hetero. The data are not contradictory each other.
Reviewer 3 Report
Comments on viruses-1667966
Title: A Nectin1 mutant mouse model is resistant to pseudorabies virus
infection
The present study by Xiaohui Yang et al. generated a genetically modified mouse model with a single site mutation, F129A, in the nectin-1 gene and evaluated the resistance of the resulting mutant mice to pseudorabies virus (PRV) infection. The work seems to be interesting and potentially useful. However, some concerns should be addressed.
Major concerns:
- The activities of the nectin-1 proteins between the mutant and wild-type mice should be compared.
- The susceptibility of the mutant mice vs wild-type mice to PRV infection should be quantified and compared using clinical and pathological scoring systems.
- The figures were poorly presented and need to be improved. Statistical analysis should be conducted to compare the death rates between the mutant and wild-type mice.
- Did all the mutant mice show microphthalmia? How about the expression of the nectin-1 protein in the mutant mice with defect in eye development?
- The manuscript should be revised by native English speakers.
Minor issues:
- Please check the author names.
- Abstract needs to be rewritten. The present study has nothing to do with genetically modified pigs!
- Keywords: delete "virus-host interaction".
- Lines 67-80, A diagram will be more informative and effective.
- Lines 107-108, The virus doses should be specified.
- Lines 121-128, procedures with specified experimental conditions or references are needed(including temperature and time).
- All the protein bands in Fig. 1B should be displayed on the same gels; Change "Viral dosage" as " Virus dose" in 2A; More mice need to be presented to show clinical signs in Fig. 2B; Fig. 3B and 3C, which is very confusing? Please clarify.
- The Discussion needs to be modified.
- The references should be kept in a consistent style as required by the Journal (e.g., references 1, 17, 30, 33, and 34).
Author Response
Major concerns:
1.The activities of the nectin-1 proteins between the mutant and wild-type mice should be compared.
Thank you for your comments. Nectin1 activity can be tested in a cell-fusion system which has been reported previously (Pertel et al., 2001; Li et al., 2017). Such system could not be easily set up for us shortly. In addition, Li’s publication has reported the cell fusion function of F129A mutant nectin1, which shows F129A reduces cell fusion compared to WT nectin1 (Li et al., 2017).
2.The susceptibility of the mutant mice vs wild-type mice to PRV infection should be quantified and compared using clinical and pathological scoring systems.
Thank you for your valuable comments. We have added the clinical score data for the viral challenge experiments in Figure S2 and S2 in the revised manuscript. We used 4 level scoring system to quantify the infection levels, which has been described in material and method section.
3.The figures were poorly presented and need to be improved. Statistical analysis should be conducted to compare the death rates between the mutant and wild-type mice.
We have added statistical analysis for the death rate results. P values were included in the revised figures 2 and 3.
4.Did all the mutant mice show microphthalmia? How about the expression of the nectin-1 protein in the mutant mice with defect in eye development?
Yes. All homozygous mutant mice show microphthalmia. We have added the western blot results showing nectin1 protein in eyes of homo and WT mice (Fig. 4D). We found nectin1 protein level in eyes of homozygous mutant mice was greatly less than that in WT littermates, however, nectin1 protein level did not differ greatly in brains between them. Severe eye abnormality may be partly explained by specifically reduced nectin1 expression in eyes in homozygous mutant mice.
5.The manuscript should be revised by native English speakers.
The manuscript has been revised by native English speakers. I believe the language has been improved greatly after revision.
Minor issues:
1.Please check the author names.
Thank you. We have checked the author names and corrected the typos.
2.Abstract needs to be rewritten. The present study has nothing to do with genetically modified pigs!
We have rewritten the abstract to describe the study more clearly and correctly.
3.Keywords: delete "virus-host interaction".
This keyword is deleted.
4.Lines 67-80, A diagram will be more informative and effective.
Thank you. We added an diagram in figure 1A describing the gene targeting strategy of mutant mice.
5.Lines 107-108, The virus doses should be specified.
The virus doses were added (105.43 and 107.42 TCID50 for 2 viral challenge experiments).
6.Lines 121-128, procedures with specified experimental conditions or references are needed(including temperature and time).
We added more text to clearly describe the experiment details.
7.All the protein bands in Fig. 1B should be displayed on the same gels; Change "Viral dosage" as " Virus dose" in 2A; More mice need to be presented to show clinical signs in Fig. 2B; Fig. 3B and 3C, which is very confusing? Please clarify.
1) Thank you for your comments. For WB in Figure 1B, the membrane was cropped to two parts according to protein molecular weight to incubate with nectin1 and actin antibodies, respectively. Therefore, they belong to the same gel. We have uploaded the original uncropped images in the system for your check.
2) "Viral dosage" are revised to "Virus dose" in the revised manuscript.
3) We have all mice pictures with different disease symptoms, but more mouse pictures seem redundant. To clearly show all symptoms of experimental mice, we added Figures S2 and S3 in the revised manuscript to include all clinical score data. 4) Figure 3B and 3C showed homo mice is resistance to PRV, but hetero and WT are similarly susceptible to PRV infection.
8.The Discussion needs to be modified.
We have modified and reorganized the discussion section. I believe the revised text will be more logic and clear to discuss and explain this study.
9.The references should be kept in a consistent style as required by the Journal (e.g., references 1, 17, 30, 33, and 34).
All references are checked and revised to be consistent with required format.
Round 2
Reviewer 2 Report
- To prove this conclusion, authors should provide whether backbone mice are susceptible to PRV or not. Which backbone mice authors used and prove they are susceptible. Is it C57BL/6?
Yes. The backbone of tested mice are C57BL/6. C57BL/6 is lethally susceptible to PRV infection. Some previous reports have clearly studied this. We have added the background information of mice and some references in Sections 2.1 and 3.2 in the revised manuscript.
## Acceptable
- How we either conclude or correlate these results with PRV infection in pigs?
Mouse models provide useful references for pigs infection with PRV. PRV can lethally infect many animals with a broad host range, implying that PRV can engage the host factors including nectin1 from different species for effective viral infection. To address the reviewers’ concern, (1) we added a sequence alignment of nectin1 of mice and pigs to show their conservation (Figure S1); (2) we added some references reporting using mice to establish PRV resistant animal models (references 34 and 35); (3) we added a paragraph in Discussion section to discuss the correlation of mice and pigs in PRV infection (Section 4.2).
## Some points are acceptable, but still questioning that for the viruses infection many factors are involved. Authors concluded nectin 1 should be critical for PRV infection in mice model but still better confirm in pig model to confirm. Even if they are similar in sequence but still other factors in other hosts should be concerned.
- How much similarity of nectin 1 protein between mice and pigs?
They have a high similarity with 93.8% homology in protein sequence. We added the nectin1 sequence alignment results among multiple species in Figure S1.
## Acceptable
- In Fig 1, authors shown very big difference in body weight. One amino acid difference between homo and hetero induced very big difference in body weight, what could be the reason?
Yes. One amino acid substitution in nectin1 can result in significant abnormality in animals. The major abnormality is eye defect in homozygous mutant mice. In addition, such mutation may impair other tissues as nectin1 works to adhere cell to cell. We have studied the eye histology of mutant mice in section 3.3. The functional impairment of nectin1 by F129A mutation thus results in the slightly reduced body weight gain of homozygous mutant mice.
## ## Acceptable
- In Fig 2, Homo mice were more resistant than WT mice which is different results from results on Fig 1. How you can explain?
Our major conclusion is that homo mice is resistant to PRV. This result is consistent throughout the manuscript. Figure 1 shows the growth status of mice, not the disease infection result. The homo mice grow up slightly slowly, but they resist PRV infection.
- In Fig 3, viral loads on homo and hetero were more or less same but very big different in survival rate. What could be the reason?
Thank you for your comments. Probably the figure is not marked clearly enough. The WT and hetero are similar in viral load and survival rate. Homo mice has low viral load and high survival rate, compared to WT and hetero. The data are not contradictory each other.
Author Response
## Some points are acceptable, but still questioning that for the viruses infection many factors are involved. Authors concluded nectin 1 should be critical for PRV infection in mice model but still better confirm in pig model to confirm. Even if they are similar in sequence but still other factors in other hosts should be concerned.
Response: Although the alphaherpesviruses have their preference to utilize different host factors for infections in their hosts, nectin1 has long been proven to be a common host factor for cell entry of most alphaherpesviruses (Geraghty, Science, 1998). In accordance with the prevailing view, our previous work also confirms nectin1 is important for PRV infection in pig cells. Knockout of nectin1 in pig cells severely impairs PRV growth (Huang, Archives of Virology, 2020). The currently submitted work further confirms the critical role of nectin1 for PRV infection in both natural and non-natural hosts. To address the reviewer’s concern, we added the contents in discussion section to discuss it as follows:
“As a common coreceptor for multiple human and animal alphaherpesviruses, nectin1 may serve a similar role for alphaherpesvirus infection in both natural and non-natural hosts [16]. Therefore, the mice model of PRV infection could be a valuable reference to study virus–host interaction in pigs.”
References:
Geraghty, R.J.; Krummenacher, C.; Cohen, G.H.; Eisenberg, R.J.; Spear, P.G. Entry of alphaherpesviruses mediated by poliovirus receptor-related protein 1 and poliovirus receptor. Science 1998, 280(5369), 1618-1620.
Huang, Y.; Li, Z.; Song, C.; Wu, Z.; Yang, H. Resistance to pseudorabies virus by knockout of nectin1/2 in pig cells. Arch. Virol. 2020, 165(12), 2837-2846.
Reviewer 3 Report
The revised manuscript has been improved significantly but needs thorough modification according to the reviewers' comments and extensive language editing.
Author Response
The revised manuscript has been improved significantly but needs thorough modification according to the reviewers' comments and extensive language editing.
Response: Thank you for your valuable comments. The whole manuscript has been carefully modified according to the reviewers’ comment. The language has been comprehensively edited by the native English speakers in the first revised version. I believe the quality of the current version of manuscript has a great improvement and can meet the requirements for publication of the journal.